# Causal associations between human plasma proteins and prostate cancer identified by proteome-wide Mendelian randomization

Lin Chen[1], Yanlun Gu[1,2], Yuke Chen[3], Wei Yu[3], Ying Zhou[1]*, Zhuona Rong[1]*, Xiaocong Pang[1]*

[1]Department of Pharmacy, Peking University First Hospital, Beijing, China; [2]School of Pharmaceutical Sciences, Peking University, Beijing, China; [3]Department of Urology, Peking University First Hospital, Beijing, China

*For correspondence:
zhouying0321@126.com (YZ);
rongzhuona@163.com (ZR);
pangxiaocong1227@163.com
(XP)

Competing interest: The authors declare that no competing interests exist.

## eLife Assessment

This study presents a meta-analysis of two independent genome-wide association studies (GWAS) that investigate the role of plasma proteins as potential biomarkers for enhancing the early detection of prostate cancer (PCa). The results provide **useful** confirmatory data that support existing evidence currently published. The evidence is **incomplete**: the study does not provide a comprehensive synthesis of all currently published work, does not explore other clinical outcomes related to prostatic disease, and its findings have not been validated through an external cohort study. These shortcomings notwithstanding, the work may be of interest to researchers studying correlates and predictors of prostate cancer risk.

**Abstract** Prostate cancer (PCa) diagnosis is hampered by the limited specificity of current methods, necessitating more reliable biomarkers. To identify causal protein biomarkers and therapeutic targets in humans, we conducted a proteome-wide Mendelian randomization (MR) study. We first performed a meta-analysis of two independent genome-wide association studies, including 94,397 individuals with PCa and 192,372 controls, which identified five possible susceptibility loci (JAZF1, PDILM5, WDPCP, EEFSEC, TNS3) for PCa. Subsequently, MR and colocalization analyses were performed using genetic instruments for 4907 plasma proteins from deCODE Genetics (N=35,559) and 2940 plasma proteins from UK Biobank Pharma Proteomics Project (UKB-PPP) (N=54,219). Among 3722 human proteins analyzed, 193 were associated with PCa risk, with 20 high-risk proteins (including KLK3) validated across both cohorts. Functional annotation implicated immune and inflammatory responses and cell–cell interaction pathways. Druggability analyses nominated several potential drug targets for PCa, such as HSPB1, RRM2B, and PSCA. Our findings reveal novel risk loci and candidate protein biomarkers, providing new etiological insights and potential avenues for PCa early detection and therapy.

## Introduction

Prostate cancer (PCa) is the second most common malignancy with an estimation of 1.5 million new cases in 2022, which is accounted for 14% of total cancer diagnosed in men worldwide (*Ferlay et al., 2024*). Even more worrying is that the number of new cases of PCa is projected to 2.9 million by 2040 (*James et al., 2024*). Despite advancements in medical technology, early detection of PCa remains

a significant challenge due to the limitations in specificity of current diagnostic methods, such as prostate-specific antigen (PSA) testing (*Kalavacherla et al., 2023*; *Tikkinen et al., 2018*). Therefore, there is an urgent need for more accurate and reliable biomarkers that can improve early detection and prognostic evaluations of PCa.

Plasma proteins have key roles in the development and progression of PCa, such as interleukin (IL)-6 (*Deichaite et al., 2022*), insulin-like growth factor (IGF)-1, and IGF-binding protein (IGFBP)-1 (*Cao et al., 2015*). However, observational studies exploring the association between plasma proteins and PCa risk are often limited by confounding factors and selection bias, making it challenging to establish a clear causal relationship.

To overcome these limitations, Mendelian randomization (MR) offers a robust methodological approach. This approach utilizes the principle of random assortment of genes from parents to offspring, using genetic variation as an instrumental variable, which mimics the randomization process in a controlled trial, thereby minimizing the impact of reverse causality (*Davies et al., 2018*). With the recent development of proteomics technology, several large-scale proteomic studies have identified over 18,000 protein quantitative trait loci (pQTLs) covering more than 4800 proteins (*Ferkingstad et al., 2021*; *Pietzner et al., 2021*). By using MR, researchers can gain a deeper comprehension of whether particular plasma proteins hold a causal nexus with PCa, thereby potentially unraveling novel biomarkers and targets for the prevention and treatment of PCa.

In this study, we performed a meta-analysis of two genome-wide association studies (GWAS) on PCa (PRACTICAL and FinnGen) for a total sample size of 94,397 cases and 192,372 controls. Based on GWAS summary statistics data from deCODE and UKB-PPP cohorts, we further performed

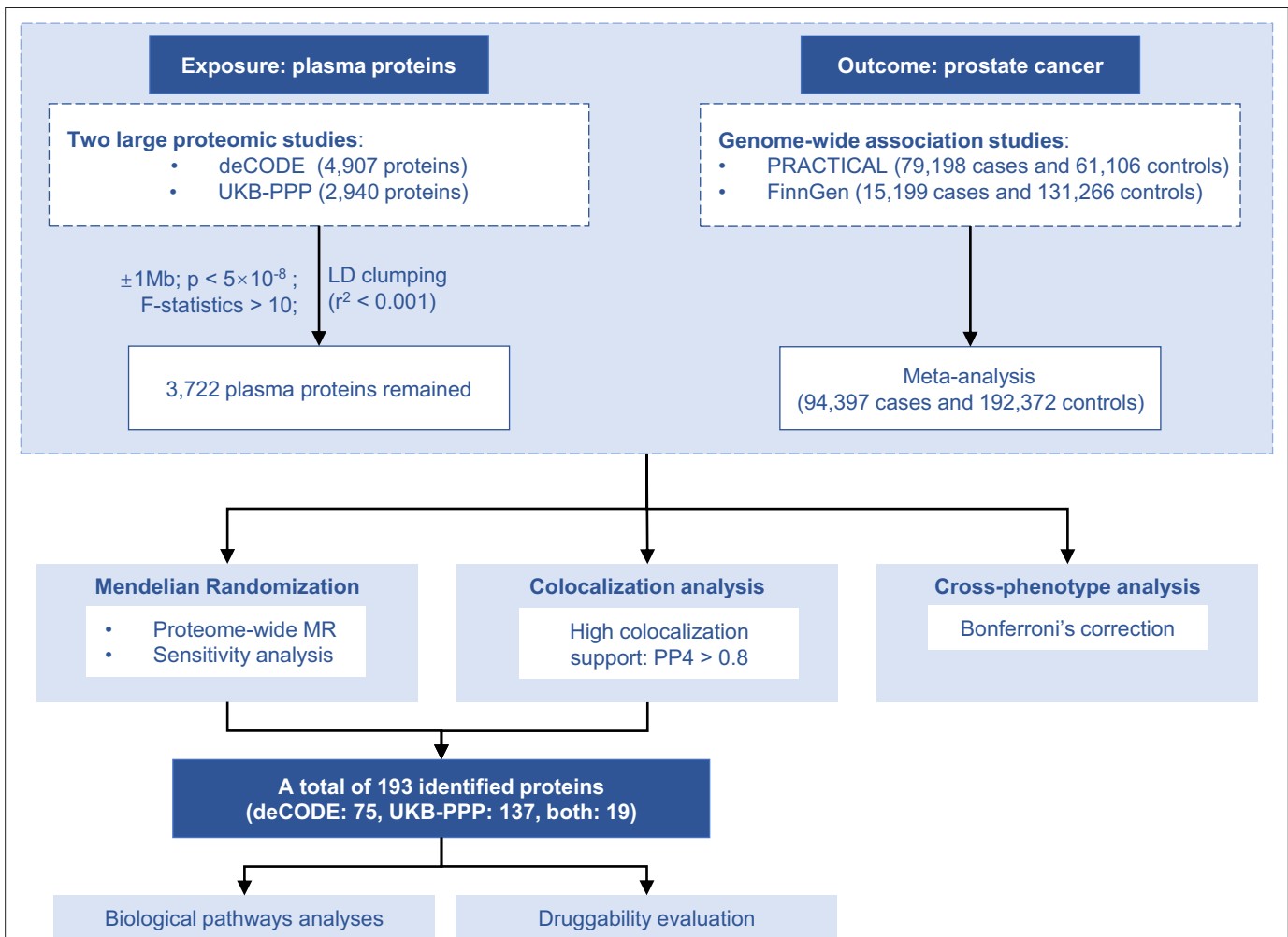

**Figure 1.** Flowchart of the study design for the protein-wide Mendelian randomization (PW-MR) analysis of prostate cancer (PCa).

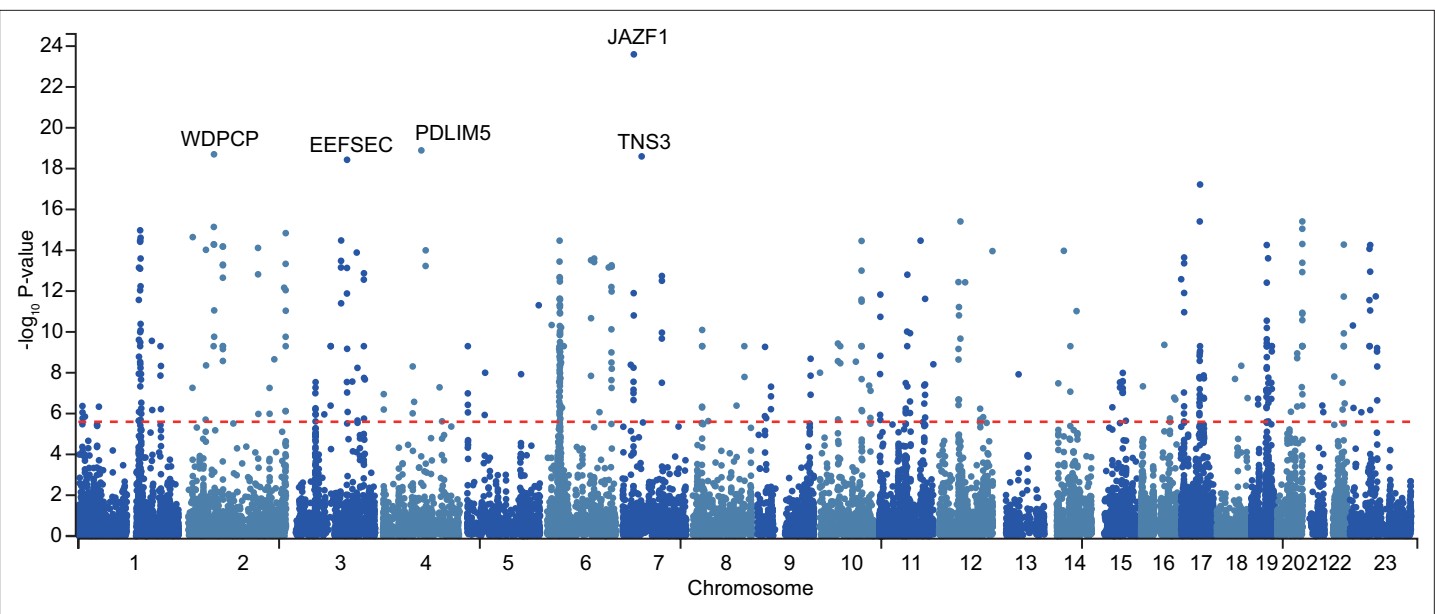

**Figure 2.** Manhattan plot of prostate cancer (PCa) genome-wide association studies (GWASs) meta-analysis. The genetic regions containing top SNPs related to PCa are depicted. The red dashed line signifies the genome-wide significance threshold of $5.0 \times 10^{-8}$.

The online version of this article includes the following figure supplement(s) for figure 2:

**Figure supplement 1.** Quantile–quantile plot of the prostate cancer (PCa) genome-wide association studies (GWASs).

**Figure supplement 2.** LocusZoom plots of genome-wide association studies (GWASs) top SNPs.

a protein-wide MR (PW-MR) study, supplemented by colocalization analysis, to explore the causal relationship between plasma proteins and PCa risk. Moreover, we indicated biological processes and pathways associated with PCa, and evaluated the druggability of risk proteins. We aimed to identify novel plasma protein biomarkers for PCa, which could address the limitations of current diagnostic methods and offer new insights into the biological mechanisms of PCa and potential therapeutic targets for intervention (*Figure 1*).

## Results

### Meta-analysis of the genome-wide association studies for prostate cancer

We conducted a meta-analysis combining two GWASs with a collective sample size of 94,397 individuals with PCa and 192,372 controls, aiming to identify genetic variants linked to PCa. The associations and assessment of Single Nucleotide Polymorphisms (SNPs) heterogeneity that passed the genome-wide *P*-value threshold at these loci with PCa are presented in *Supplementary file 1A*. We found five genetic risk loci contained at least one SNP passing the genome-wide significance threshold of $P \leq 5 \times 10^{-8}$: JAZF1, PDILM5, WDPCP, EEFSEC, and TNS3 (*Figure 2*). Among them, PDILM5, WDPCP, EEFSEC, and TNS3 were promising candidates as novel susceptibility loci associated with PCa. *Figure 2—figure supplement 1* displays the associated quantile–quantile plot. The LocusZoom plots of the top SNPs at JAZF1, PDILM5, WDPCP, EEFSEC, and TNS3, along with their genomic location, GWAS *P* values, and recombination rate with neighboring SNPs are visualized in *Figure 2—figure supplement 2*. In summary, this GWAS meta-analysis discovered genetic variations in one recognized PC-associated locus and four potential novel loci, providing a reliable dataset for MR analyses.

### Cross-phenotype analysis of prostate cancer

The cross-phenotype analysis was designed to systematically identify phenotypic traits that share genetic or molecular pathways with PCa, thereby uncovering pleiotropic mechanisms or shared risk factors. We used the iCPAGdb to conduct this analysis with PCa genome-wide significant SNPs ($P \leq 5$

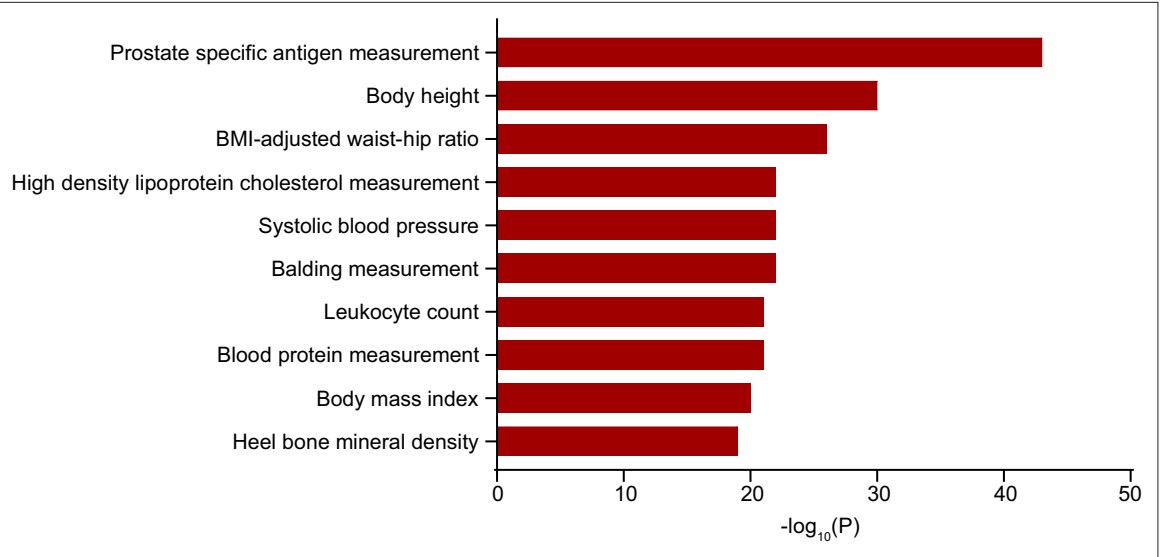

**Figure 3.** The top 10 significant cross-phenotype associations with prostate cancer (PCa) at a 5% false discovery rate (FDR). The x-axis represents the *P* value of the correlation. BMI, body mass index.

× 10⁻⁸). The iCPAGdb offered an improved algorithm for identifying cross-phenotype associations by using pre-computed ancestry-specific LD databases and integrating genetic data from 3793 traits in the NHGRI-EBI GWAS catalog (*Wang et al., 2020*). Cross-phenotype analysis of PCa-associated SNPs identified 117 traits significantly linked to PCa after Bonferroni's correction (*Supplementary file 1B*). As shown in *Figure 3*, the strong association with PSA measurement ($P=7.16 \times 10^{-43}$) validated our genetic instruments, while the enrichment of metabolic traits such as BMI-adjusted waist–hip ratio (WHR) ($P=1.32 \times 10^{-26}$), high-density lipoprotein (HDL) cholesterol measurement ($P=2.38 \times 10^{-22}$), and body mass index ($P=1.71 \times 10^{-20}$) aligned with epidemiological evidence linking obesity to aggressive PCa. Notably, the strong association with balding ($P=3.01 \times 10^{-22}$) reinforced the androgen axis in PCa pathogenesis and the correlation with heel bone mineral density ($P=2.02 \times 10^{-19}$) was consistent with the clinical observations of osteoblastic bone metastasis in advanced PCa. These cross-phenotype associations collectively highlight PCa as a systemic disorder involving androgen signaling, metabolic dysregulation, and skeletal interactions, providing novel insights into its multifactorial etiology.

### Proteome-wide Mendelian randomization studies of prostate cancer

The genetic association summary statistics of 35,559 Icelanders from deCODE Genetics and 54,219 Europeans from the UKB-PPP were utilized to investigate the relationship between PCa and plasma proteins. Our genetic instrument selection strategy enabled us to examine 1778 and 1944 proteins from deCODE and UKB-PPP (*Supplementary file 1C and D*). Using the Wald ratio or inverse-variance weighted (IVW) method, a total of 193 unique plasma proteins were significantly associated with PCa after multiple tests with a 5% false discovery rate (FDR) correction. This analysis yielded 137 proteins in UKB-PPP more than 76 proteins in deCODE (*Figure 4A*). The results of the heterogeneity test based on Q statistics showed little evidence of heterogeneity. Furthermore, no significant intercept was detected, implying that there was no directional pleiotropy observed.

After FDR correction, 20 proteins were detected in both data sets and the identified associations were consistent (*Figure 4B*). Genetic prediction indicated that 11 of these 20 proteins were positively associated with the risk of PCa, as well as the remaining 9 proteins were negatively associated, suggesting that these 9 proteins may be protective factors against PCa (*Supplementary file 1E*). A PhenoGram depicts the chromosomal location of the 193 unique identified proteins in deCODE and UKB-PPP studies (*Figure 4C*).

### Colocalization analysis

We performed colocalization analyses of proteins significantly expressed in deCODE and UKB-PPP studies with PCa. It was observed that four proteins in the deCODE study and seven proteins in the

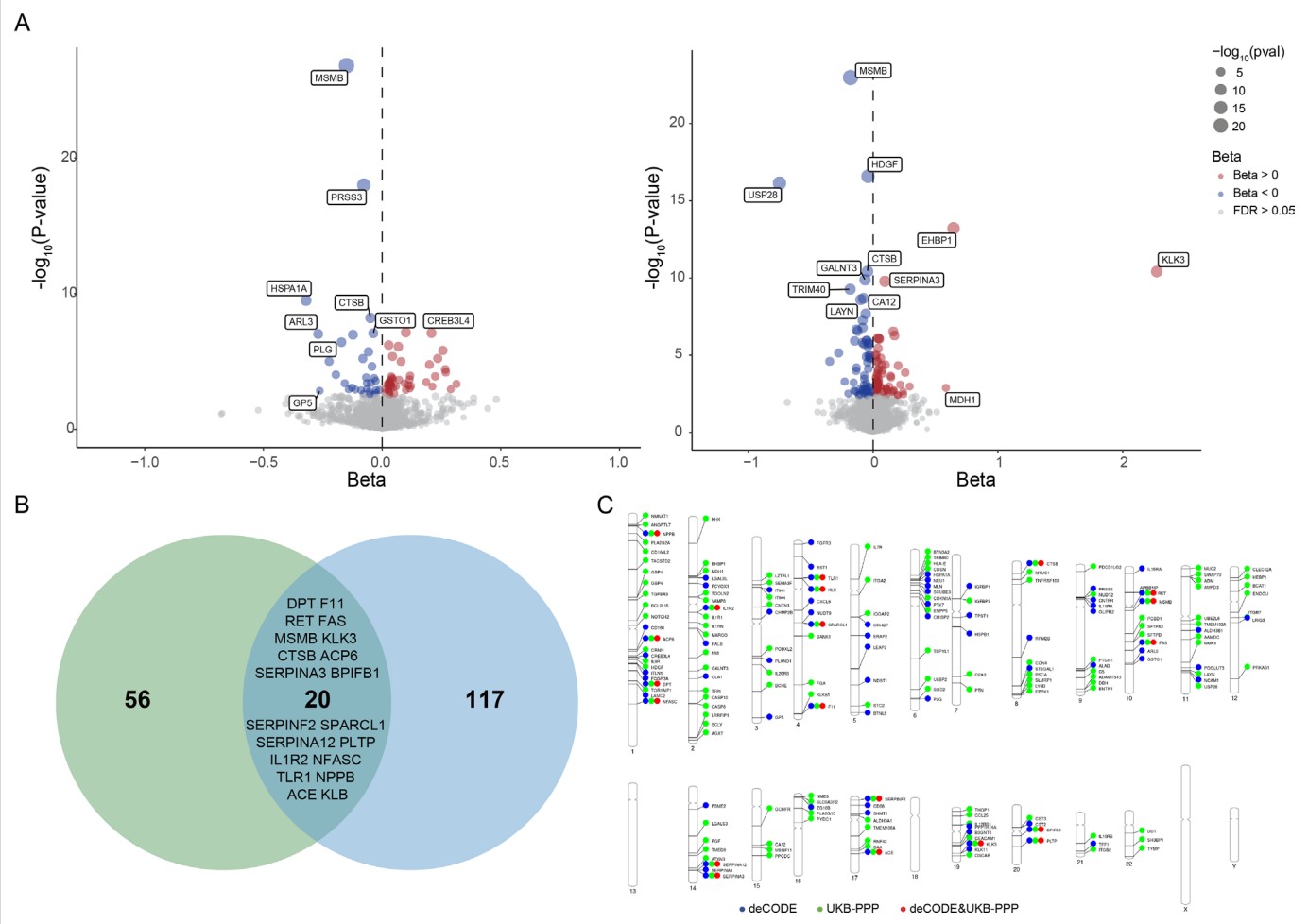

**Figure 4.** Result of protein-wide Mendelian randomization (PW-MR) on the associations between plasma proteins and the risk of prostate cancer (PCa). (**A**) Volcano plot of PCa PW-MR study using deCODE (the left side) and UKB-PPP (the right side) cohorts. Annotated proteins passed the 5% false discovery rate (FDR) IVW *P*-value threshold. The blue and red colors represent a negative and positive effect, respectively. (**B**) Venn diagram depicting proteins associated with PCa in deCODE and UKB-PPP. (**C**) PhenoGram of PCa PW-MR study significant associations. The blue dots and the green dots represent the deCODE and UKB-PPP-specific proteins, respectively. The red dot represents both simultaneously.

UKB-PPP were colocalized with PCa associations with high support of evidence (PPH4≥0.8) (*Table 1*), suggesting that these 10 plasma proteins might serve as potential targets for treating PCa. Among them, SERPINA3 showed strong colocalization evidence in both deCODE (PPH4=0.952) and UKB-PPP (PPH4=0.951) studies. This analysis identified one causal variant in deCODE (rs61976125) and one causal variant in Fenland (rs6575449) (*Figure 5*).

## Gene-based association and pathway analyses

We used the GENE2FUNC tool available in FUMA to explore the biological significance, functional implications, and tissue-specific expression of the genes identified from our GWAS. These genes appeared to be significantly enriched in inflammatory and immune pathways (such as defense, immune, and inflammatory response), as well as cell interaction and signaling pathways (such as interaction between organism and cell adhesion) (*Figure 6A*). The Kyoto Encyclopedia of Genes and Genomes KEGG0 pathway enrichment analysis revealed several pathways that were significantly enriched, such as cytokine–cytokine receptor interaction, p53 signaling pathway, JAK-STAT signaling pathway, pathways in cancer, and apoptosis (*Figure 6B*). These analyses indicated potential biological processes and mechanisms associated with PCa. The 193 unique genes were predominantly expressed in the lymphocytes, blood, liver, and prostate (*Figure 6—figure supplement 1*).

**Table 1.** Analysis of Mendelian randomization and colocalization of significant proteins with prostate cancer.

| Genome-wide association studies | Outcomes | Proteins | Mendelian randomization | | | Colocalization analysis PH4 |
| --- | --- | --- | --- | --- | --- | --- |
| | | | OR (95% CI) | P value | P value after false discovery rate adjustment | |
| deCODE | Prostate cancer | MSMB | 0.86 (0.84,0.88) | 1.56E-27 | 2.82E-24 | 0.999 |
| | | POGLUT3 | 0.89 (0.83,0.96) | 1.16E-03 | 2.44E-02 | 0.998 |
| | | PRSS3 | 0.93 (0.91,0.94) | 1.00E-18 | 6.03E-16 | 0.956 |
| | | SERPINA3 | 1.11 (1.06,1.15) | 7.18E-08 | 1.05E-05 | 0.952 |
| UKB-PPP | Prostate cancer | USP28 | 0.47 (0.40,0.56) | 7.27E-17 | 2.89E-14 | 0.999 |
| | | KLK3 | 9.70 (4.94,19.03) | 3.83E-11 | 6.91E-09 | 0.999 |
| | | IGFBP3 | 1.07 (1.05,1.09) | 2.04E-11 | 4.49E-09 | 0.997 |
| | | CASP10 | 1.22 (1.11,1.34) | 4.73E-05 | 1.54E-03 | 0.988 |
| | | HDGF | 0.96 (0.95,0.97) | 2.61E-17 | 1.29E-14 | 0.966 |
| | | SERPINA3 | 1.10 (1.07,1.13) | 1.64E-10 | 2.32E-08 | 0.951 |
| | | C5 | 1.19 (1.11,1.27) | 5.32E-07 | 4.06E-05 | 0.883 |

OR = odds ratio. CI = confidence interval.

## Druggability of identified proteins

Exploring new therapeutic opportunities for PCa based on genetic information is crucial for developing targeted treatments. In our study, we examined the druggability of genes and proteins identified in MR analyses using OpenTargets databases. We investigated 128 unique drugs targeting 45 identified proteins (*Supplementary file 1F*). Among these, three drugs (APATORSEN, TRIAPINE, and MK-4721) currently in clinical trials were intended for PCa treatment, with each targeting HSPB1, RRM2B, and PSCA, respectively. Notably, the effects of these drugs on their respective protein targets align with the directions indicated by our MR results, suggesting a consistency between genetic evidence and therapeutic potential. In addition, several other identified targets, such as RET, FGFR3, NCAM1, TYMP, TNFRSF10B, MMP3, TACSTD2, and NOTCH2, were implicated in various cancers and present potential therapeutic avenues.

## Discussion

### Genetic susceptibility loci and functional mechanisms

In this study, we conducted a comprehensive meta-analysis of two GWAS for PCa, identifying significant genetic loci associated with PCa risk. The combined sample size of 94,397 cases and 192,372 controls revealed one known (JAZF1) and four potential novel (PDLIM5, WDPCP, EEFSEC, and TNS3) susceptibility loci of PCa. These genetic findings not only expand our understanding of PCa heritability but also bridge the gap between genetic risk and downstream molecular mechanisms.

JAZF1, a transcriptional repressor linked to metabolic regulation (*Rosario et al., 2023*) and cellular proliferation (*Sung et al., 2018*), has been implicated in both type 2 diabetes and PCa risk through GWAS (*Machiela et al., 2012*; *Sánchez-Maldonado et al., 2022*). Its dual role highlights the interaction between metabolic dysregulation and oncogenesis, indicating that interventions targeting metabolic pathways may hold therapeutic promise for PCa.

PDLIM5 (PDZ and LIM domain 5), a cytoskeleton-associated protein, regulates cell migration and tumor progression by binding a variety of proteins through its specific domains. In PCa, PDLIM5 promotes epithelial–mesenchymal transition and migration of PCa cells (*Liu et al., 2017*), while downregulating the expression of PDLIM5 may ultimately impede the progression of PCa (*Xie et al., 2020*), indicating its potential value in predicting the risk of advanced PCa. WDPCP, a key effector of planar cell polarity (PCP) signaling, regulates actin cytoskeleton to maintain tissue architecture (*Cui et al., 2013*). Disruption of PCP signaling can cause abnormal cell behavior and

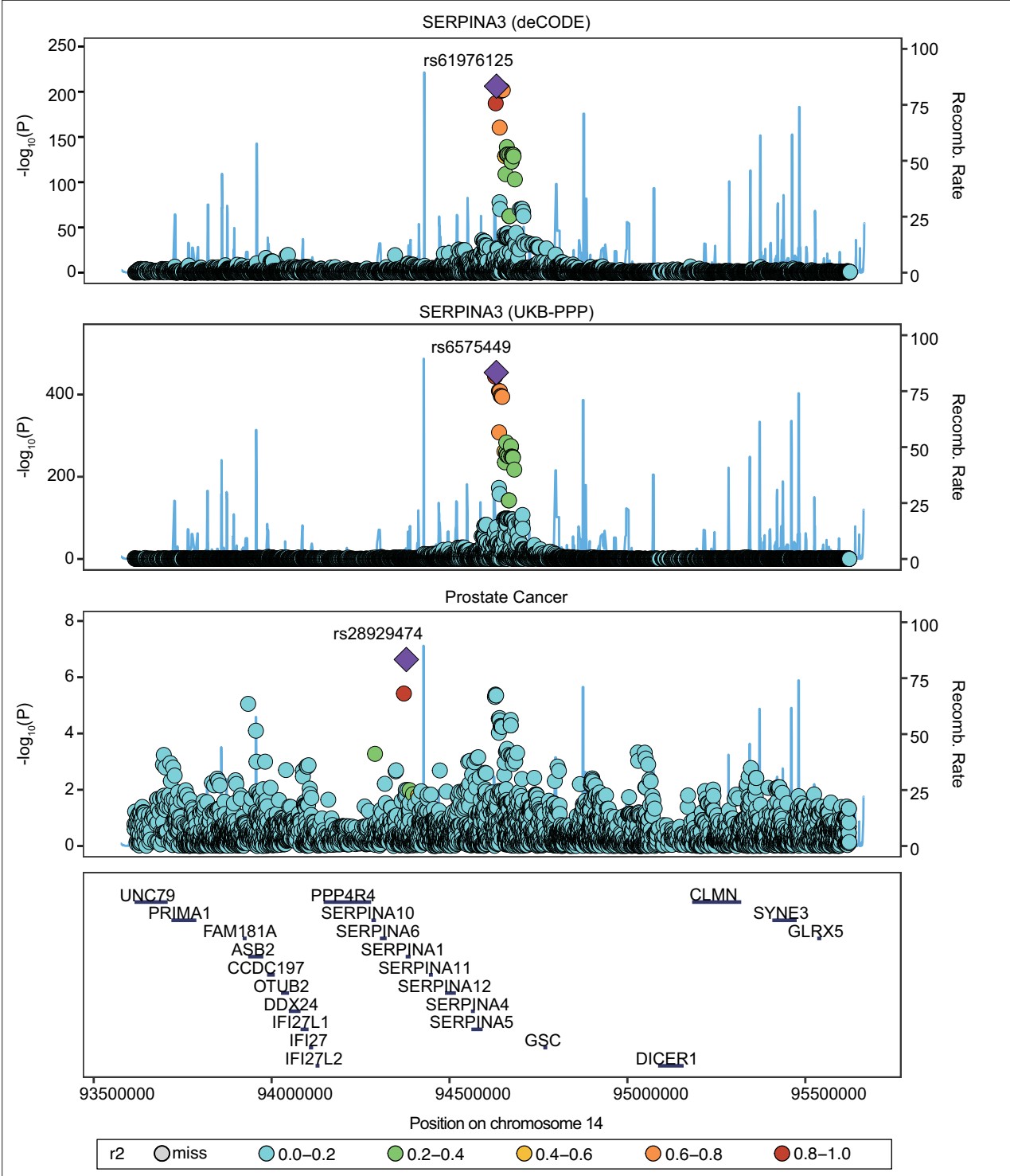

**Figure 5.** Colocalization plot of SERPINA3 variants associated with prostate cancer (PCa) in deCODE and UKB-PPP. Variants are color-coded based on their linkage disequilibrium (LD) with the lead SNP (the variant with the lowest *P*-value). The lead SNP is highlighted in red. Other variants are colored from blue to yellow, indicating decreasing LD with the lead SNP.

tissue structure, leading to the development of cancer (*Humphries and Mlodzik, 2018*). EEFSEC is a member of the eukaryotic family of elongation factors, and its role in tumor cells has been rarely reported. A functional study demonstrated that EEFSEC was significantly upregulated in PCa cells, and high expression of EEFSEC was associated with poor prognosis in patients with PCa

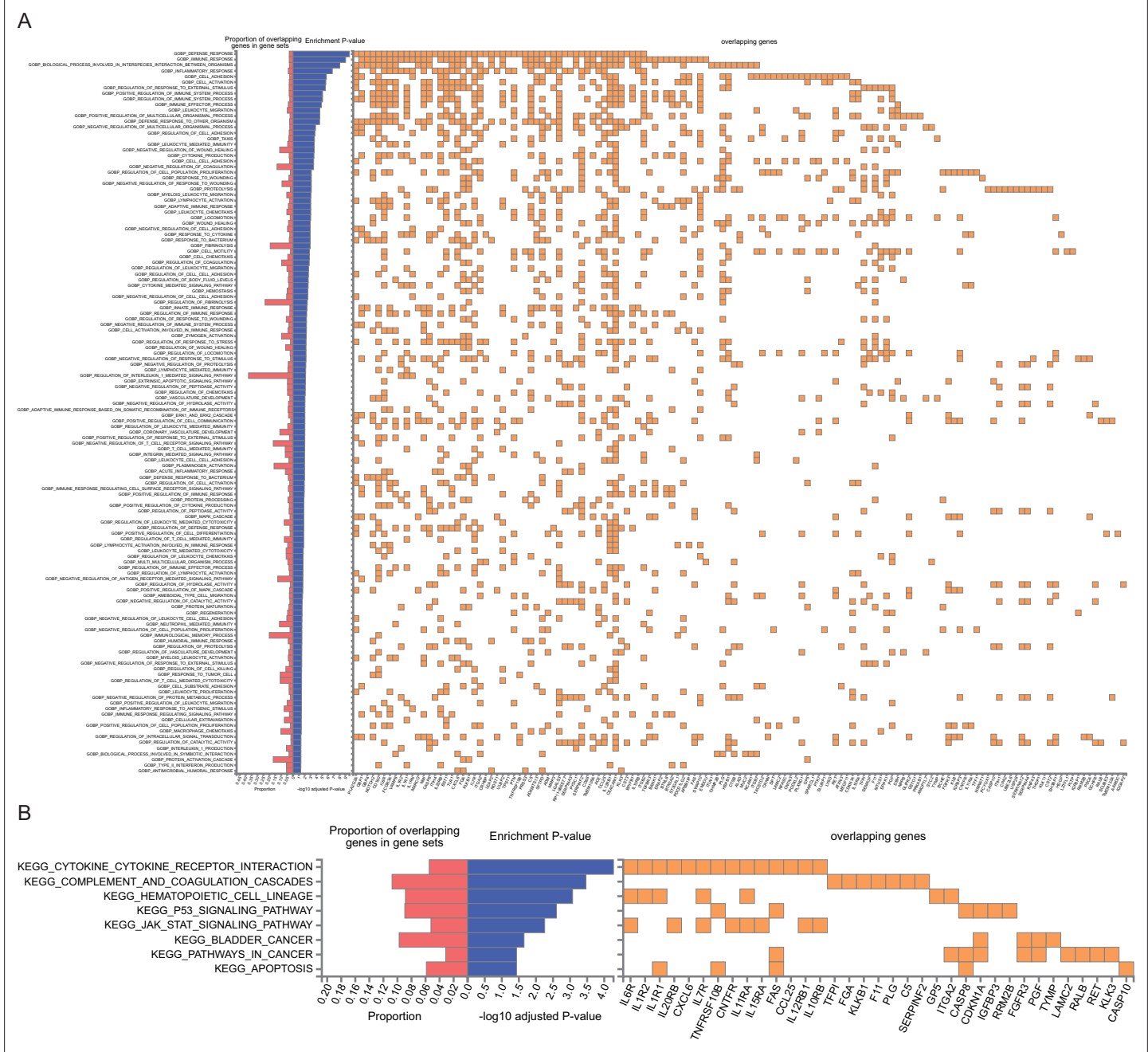

**Figure 6.** Functional annotation of the genetic architecture of prostate cancer (PCa). (**A**) Biological processes and (**B**) KEGG pathway analysis of the 193 unique proteins identified in deCODE and UKB-PPP.

The online version of this article includes the following figure supplement(s) for figure 6:

**Figure supplement 1.** Gene expression heat map of the 193 unique protein-wide Mendelian randomization (PW-MR) significant proteins in deCODE and UKB-PPP.

(*Xu et al., 2021*). TNS3 promotes cell migration and invasion through focal adhesion dynamics (*Chen et al., 2017*; *Martuszewska et al., 2009*; *Zheng et al., 2021*; *Zuidema et al., 2022*), which is critical in cancer metastasis. While these loci implicate diverse pathways, their collective impact may converge on tumor microenvironment (TME) reprogramming. To determine whether these four genes are causative factors for PCa, genetic association studies with a larger sample size will be needed.

## Multifactorial drivers of prostate cancer

PCa arises from a complex interplay of genetic, metabolic, and hormonal perturbations. In the cross-phenotype analysis of PCa, we found multiple risk-associated traits, including proteomic biomarkers (PSA), metabolic dysregulation (central obesity, HDL dysfunction, hypertension), and androgen signaling (balding), collectively underscoring PCa's multifactorial etiology.

PSA, a protein secreted by both normal and malignant prostate epithelial cells, has long been a cornerstone in the early detection and monitoring of PCa. However, the specificity and sensitivity of PSA as a diagnostic tool have been questioned due to its elevation in benign conditions such as prostatitis and benign prostatic hyperplasia (BPH), leading to unnecessary biopsies and substantial overdiagnosis (*Gudmundsson et al., 2018*; *Ilic et al., 2018*; *Kalavacherla et al., 2023*). Therefore, complementary biomarkers are urgently needed to improve diagnostic accuracy. Our PW-MR study revealed that genetically plasma MSMB levels were causally associated with PCa risk. Importantly, this aligns with the clinical utility of urinary MSMB: a quadriplex urine panel (MSMB, TRPM8, AMACR, PCA3) improved diagnostic accuracy in PCa patients compared to PSA alone (*Jamaspishvili et al., 2011*). While plasma MSMB reflects systemic risk, urinary MSMB captures localized prostate pathology, together offering complementary insights into PCa biology. In conclusion, systematic integration of proteomic biomarkers identified through causal inference frameworks with PSA-based screening offers a path to resolve the persistent challenges of PCa diagnostic accuracy.

Emerging evidence implicates metabolic syndrome (MetS), which encompasses a cluster of conditions, including central obesity, dyslipidemia, and hypertension, may be a risk factor of PCa and may also worsen outcomes (*Hernández-Pérez et al., 2022*; *Lifshitz et al., 2021*). The significant association between BMI-adjusted WHR and advanced PCa risk suggested that central obesity may play a critical role in the development of PCa (*Genkinger et al., 2020*; *Perez-Cornago et al., 2022*). Visceral adiposity may increase risk of advanced forms of PCa through changes in cytokines and growth factors, hormone regulation, and metabolism (*Doyle et al., 2012*; *Himbert et al., 2017*). These metabolic perturbations extend beyond obesity itself, intersecting with broader lipid and inflammatory pathways that further modulate PCa risk. For instance, HDL cholesterol, while classically recognized for its cardioprotective roles in reverse cholesterol transport and antioxidant and anti-inflammatory activities (*Rohatgi et al., 2021*), may also antagonize prostate carcinogenesis. Epidemiological and preclinical studies found that HDL may suppress prostate carcinogenesis by reducing oxidative stress and the levels of pro-inflammatory molecules in cancer cells and TME (*Ossoli et al., 2022*; *Ruscica et al., 2018*). However, in obesity, HDL's antioxidant and anti-inflammatory capacity is impaired due to altered composition and reduced functionality (*Bacchetti et al., 2024*), while adipose-derived ROS and cytokines further amplify oxidative damage (*Balan et al., 2024*). The loss of HDL protection and obesity-driven oxidative stress provides a physiological mechanistic basis for obesity-associated PCa aggressivity. This metabolic-inflammatory axis extends to hypertension, which is another MetS hallmark. Although results from previous studies of the association between hypertension and PCa development remain inconsistent (*Christakoudi et al., 2020*; *Liang et al., 2016*; *Seretis et al., 2019*), an MR analysis suggested that elevated systolic blood pressure might increase PCa risk through systemic inflammation (*Stikbakke et al., 2022*), a mechanism similar to that of central obesity and HDL dysfunction. Chronic inflammation contributes to a pro-tumorigenic environment by promoting cellular proliferation, DNA damage, and resistance to apoptosis (*Mantovani et al., 2008*). Thus, MetS components likely converge on overlapping pathways to accelerate PCa progression, though the precise relationship between systolic blood pressure and PCa remains unclear. More research is needed to elucidate the exact biological pathways involved and to determine whether managing clinical features of MetS could serve as a preventive strategy for PCa.

The link between androgenic alopecia (AGA) and PCa risk further implicates androgen metabolism as a central driver, as both conditions are influenced by dihydrotestosterone (DHT) levels. In AGA, DHT binds to androgen receptors (AR) in the dermal papilla cells, causing follicular miniaturization and hair loss (*Urysiak-Czubatka et al., 2014*). Similarly, DHT-driven AR activation in prostate epithelial cells promotes proliferation and inhibits apoptosis, fostering tumorigenesis (*Tong et al., 2022*). However, the link between AGA and PCa may not be solely hormone-dependent. Genetic pleiotropy, where variants in loci such as AR or SRD5A2 influence both balding and prostate carcinogenesis, could partially explain this association (*Hayes et al., 2005*; *Hayes et al., 2007*). Further studies are needed to confirm the causality.

Overall, our cross-phenotype analysis provided a comprehensive view of the diverse factors associated with PCa and enhanced our understanding of the disease's multifaceted nature. Future research should focus on elucidating the underlying mechanisms of these associations and exploring their potential for integration into clinical practice.

## Plasma proteins causal links to prostate cancer

The PW-MR study supplemented by colocalization analysis identified 193 unique proteins causally associated with PCa, including 10 with robust colocalization evidence. These findings pinpoint both established and novel protein mediators of PCa pathogenesis, offering actionable insights into diagnostic and therapeutic innovation.

Microseminoprotein-beta (MSMB), the second most abundant prostate-secreted protein after PSA (*Lilja and Abrahamsson, 1988*), has been shown to control prostate cell growth by regulating apoptosis (*Garde et al., 1999*). Unlike PSA, MSMB expression is not directly regulated by androgens. This androgen independence makes it possible for MSMB to be a supplementary biomarker for PSA, particularly in contexts where androgen receptor signaling is dysregulated, such as castration-resistant prostate cancer (CRPC) (*Sjöblom et al., 2016*). Our study found that MSMB demonstrated the strongest protective association, and this finding was validated in prospective studies (*Smith Byrne et al., 2019*; *Haiman et al., 2013*), supporting a potential protective role of MSMB in PCa. Some studies reported the level of MSMB in benign prostate tissues or BPH was significantly elevated and decreased or lost in PCa (*Luebke et al., 2019*; *Nam et al., 2006*; *Whitaker et al., 2010*); however, others paradoxically observed that the decreased expression of MSMB was both in the tumor (especially in more advanced tumor) and adjacent benign prostate tissue (*Bergström et al., 2018*). This apparent discrepancy may reflect dynamic compensatory regulation during disease progression. In early carcinogenesis, benign epithelia adjacent to low-grade tumors may upregulate MSMB as a protective response to counteract oncogenic stress, whereas advanced tumors drive epigenetic silencing of MSMB, propagating a permissive microenvironment for invasion. Overall, these findings position MSMB as a promising biomarker, which is expected to enhance the diagnostic and prognostic specificity of PCa through integration with PSA.

SERPINA3 is an inhibitor of serine proteases that was previously found to be upregulated in various types of cancer and its elevation was associated with a worse prognosis (*de Mezer et al., 2023*). Mechanistically, SERPINA3 is regulated by inflammatory cytokines so that its expression is increased in the inflammatory response (*Péré-Brissaud et al., 2015*) and may drive tumor progression. SERPINA3 emerged as a significant risk factor for PCa in our MR analysis, which aligns with several proteomics (*Nguyen et al., 2018*; *Zhang et al., 2022*). The overexpression of SERPINA3 indicates that damage tends to occur in the body contributing to decreased cell adhesion ability and inhibition of apoptosis (*Chelbi et al., 2012*). These alterations may synergistically enhance metastatic potential, as evidenced by a recent study linking SERPINA3 to bone metastasis. A study identified that the enhanced expression of SERPINA3 stimulated the bone environment by promoting osteoblasts and osteoclasts activation, suggesting that it may serve a diagnostic biomarker for PCa with bone metastasis phenotype and survival (*Ito et al., 2023*).

PRSS3, also named mesotrypsin, is reported to be aberrantly expressed in various types of tumors and participates in the progression and development of cancers. For example, PRSS3 was identified to be downregulated in lung cancer (*Zhou et al., 2023*) but upregulated in pancreatic cancer (*Jiang et al., 2010*; *Xing et al., 2019*), suggesting that it may have different roles depending on the cellular or disease ways. A study demonstrated that PRSS3 was ectopically expressed in metastatic tissues of PCa and identified PRSS3 as a promising therapeutic target for metastatic PCa (*Hockla et al., 2012*). The research of PRSS3 in PCa is relatively limited, and further studies are necessary to fully understand the functions and mechanisms of PRSS3 in human PCa.

KLK3, which codes for PSA, is a well-known PCa biomarker. While its diagnostic utility is clinically validated, emerging evidences reveal that KLK3 mRNA detection in whole blood has a predictive role in tumor progression and therapeutic resistance, which is expected to become a potential prognostic marker for PCa (*Boerrigter et al., 2021*; *Cho et al., 2024*). KLK3's canonical role as a serine protease involves cleaving extracellular matrix components and activating matrix metalloproteinases to promote tumor invasion and metastasis (*Escaff et al., 2010*; *Moradi et al., 2019*). Additionally, the androgen regulation of KLK3 expression correlated with the androgen receptor signaling pathway, a

critical driver of PCa progression, particularly in CRPC (*Lilja et al., 2008*). Despite these insights, the precise biological pathways through which KLK3 contributes to PCa progression remain incompletely understood. Further research is needed to illustrate the direct and indirect effects of KLK3 on PCa cells and TME.

Despite substantial preclinical efforts identifying numerous promising biomarkers, their translation into routine clinical application remains strikingly limited due to the lack of further validation. Therefore, prospective validation is important and crucial.

## Biological mechanisms and druggable targets

Our integrative analysis revealed that the 193 candidate genes identified through MR were significantly enriched in inflammatory/immune pathways (e.g., cytokine–cytokine receptor interaction, JAK-STAT signaling) and cancer-related processes such as apoptosis and cell adhesion. These pathways not only depict the molecular landscape of PCa but also identify viable targets for therapeutic intervention.

For instance, RRM2B is a DNA repair enzyme targeted by the clinical-stage drug TRIAPINE, and its role in maintaining genomic stability under replicative stress is consistent with the enrichment of the p53 signaling pathway (*Elfar et al., 2024*). Similarly, the cell adhesion pathways are central to metastatic dissemination. As a member of the Thy-1/Ly-6 family involved in cell adhesion and metastasis, PSCA's overexpression in high-grade PCa and metastatic lesions underscores the therapeutic potential of PSCA inhibitors in targeting tumor-stroma crosstalk to prevent metastatic spread (*Nayerpour Dizaj et al., 2024*). Furthermore, HSPB1, a molecular chaperone implicated in apoptosis evasion and chemotherapy resistance (*Garrido et al., 2006*; *Hadaschik et al., 2008*; *Kamada et al., 2007*), aligns with the observed enrichment of survival pathways in aggressive PCa (*Shiota et al., 2013*; *Vasiljević et al., 2013*). The antisense oligonucleotide APATORSEN, which silences HSPB1, restores chemosensitivity by reactivating apoptotic signaling, offering a strategy for therapeutic resistance against CRPC (*Le et al., 2023*). Beyond these prioritized targets, the druggability analysis uncovered broader therapeutic potential. Proteins like RET (enriched in JAK-STAT signaling) and FGFR3 (linked to pathways in cancer) are already targeted by approved therapies in other malignancies, suggesting rapid repurposing potential for PCa. In conclusion, bridging pathway enrichment with druggability data transforms genetic associations into therapeutic hypotheses.

## Study strengths and limitations

This study possessed several strengths, including the use of the largest collection of plasma protein data (covering more than 4800 proteins), large sample sizes of GWAS, a mutual validation across two independent outcome datasets, and the use of colocalization analysis to support the MR results. Additionally, our assessment of the human blood proteome relied on two technologies (SOMAmer and Olink), which were valuable for identifying plasma proteins associated with disease traits such as PCa.

Some limitations of our analysis should be acknowledged. Firstly, this investigation was only confined to Europeans, restricting the applicability of our findings to other populations. It is crucial for future studies to identify risk proteins in more diverse populations, especially African ancestry who face a higher risk of PCa. Secondly, our study concentrated on the proteins available in the MR analysis, which likely led to the omission of other potential therapeutic targets. Thirdly, we utilized proteomic data from Icelanders whose genetic backgrounds may differ from other European populations, which might introduce bias. Lastly, our analysis relied exclusively on publicly available GWAS summary statistics from openGWAS and FinnGen, which did not provide individual-level data on covariates, resulting in no direct assessment of demographic or clinical differences between cases and controls. However, this potential bias might be minimal, as we found that 20 proteins showing significance ($P<0.05$) in both the deCODE and UKB-PPP studies had consistent associations with PCa.

In conclusion, we conducted a large-scale PW-MR study using the MR and colocalization analysis to investigate the genetic associations of up to 3722 unique proteins with PCa. We revealed the complex genetic architecture of PCa and identified many new plasma proteins with strong causal associations to PCa. These findings highlighted the potential biomarkers for early detection and therapeutic targets, providing a foundation for future research and potential clinical applications.

# Materials and methods

## Study design

An overview of the analytical framework is depicted in *Figure 1*. Briefly, the proteome-wide MR study was designed to assess the causal effects of human plasma proteins on PCa risk. We integrated genetic instruments for 3722 plasma proteins from two large-scale studies with summary-level genome-wide association data from a meta-analysis of 94,397 cases and 192,372 controls. The analytical workflow comprised protein-specific MR estimates, sensitivity, and colocalization analyses to verify robustness, followed by cross-phenotype, biological pathway, and druggability evaluations of the significant findings.

## Data sources for plasma proteins

We selected *cis*-SNPs associated with plasma proteins as instrumental variables from two large-scale GWASs in the deCODE Genetics (*Ferkingstad et al., 2021*) and UKB-PPP (*Sun et al., 2023*). *cis*-SNPs were defined as SNPs within a vicinity of ±1 Mb around the gene encoding the protein. From these SNPs, only those with a minor allele frequency of ≥1% that were genome-wide significance ($P<5 \times 10^{-8}$) and considered independent (linkage disequilibrium $r^2$ <0.1 in 1000G) were retained. deCODE Genetics conducted proteomic profiling on blood plasma samples from 35,559 Icelanders using the SomaScan platform and collected data on 4907 aptamers (*Ferkingstad et al., 2021*). For the two-sample MR analysis, we selected *cis*-SNPs as instrumental variables for 1778 proteins. Likewise, *cis*-SNPs for 1944 plasma proteins were obtained from the UKB-PPP where 2940 proteins were measured among 54,219 Europeans using the Olink platform (*Sun et al., 2023*). The proteins with positive MR results were included in the colocalization analysis.

## Data sources for prostate cancer

GWAS summary statistics for PCa were obtained from the Prostate Cancer Association Group to Investigate Cancer Associated Alterations in the Genome (PRACTICAL) consortium (*Schumacher et al., 2019*) and FinnGen study. The PRACTICAL consortium included 79,198 cases and 61,106 controls. We used the data on PCa from the FinnGen study R10 in this analysis, which comprised 15,199 cases and 131,266 controls. All participants of these two cohorts were of European ancestry. In the MR analysis, we treated the PRACTICAL consortium as the discovery study and the FinnGen R10 study as the replication. To increase the effectiveness, we performed a fixed-effect GWAS meta-analysis of the two GWASs using the METAL package. The quantile–quantile plot was generated using the 'qqman' package in R software (4.3.3). The R package 'gassocplot' was used to plot regional association plots for the top SNP at each of the genome-wide significant loci identified.

## Cross-phenotype analysis

Through the interactive cross-phenotype analysis of GWAS database (iCPAGdb) (*Wang et al., 2020*), a new platform for cross-phenotype analysis, we explored the genetic correlation between PCa and other traits. We analyzed the genetic correlations between PCa and 3793 traits using data from the National Human Genome Research Institute-European Bioinformatics Institute (NHGRI-EBI) GWAS catalog (*Wang et al., 2020*). Through the use of ancestry LD-specific association data, iCPAGdb performs cross-phenotype enrichment analyses. iCPAGdb reveals signals of pairwise traits and shared signals by analyzing traits associations with LD proxy SNPs. The output data will show results from Fisher's exact test with adjustment for 5% FDR and also Bonferroni's, Jaccard's, Sorensen's, and Chao–Sorensen similarity indexes.

## MR analysis

We conducted a MR analysis with plasma proteins as the exposure variable and PC as the outcome variable. We utilized the R package 'TwoSampleMR' in R software (4.3.3) for the MR analysis. When only one SNP was available for a particular protein, we applied the Wald ratio test. IVW was used as the main analysis method for two or more SNPs available. Considering multiple testing, we employed the 5% FDR method for *P*-value correction. $P<0.05$ was considered statistically significant. MR-Egger and weighted median methods were used as supplementary analysis methods. Cochran's Q statistic test using the MR-Egger method was used to determine the heterogeneity between the genetic

variants. We also performed MR-Egger regression intercept to detect and adjust the directional horizontal pleiotropy (*Hemani et al., 2018*). $P<0.05$ was considered the presence of directional pleiotropy and thus removed from the further analyses.

## Colocalization analysis

The 'coloc' package was used to perform Bayesian colocalization analysis to investigate if the associations between plasma proteins and PCa were driven by linkage disequilibrium (LD) (*Giambartolomei et al., 2014*). For proteins with positive MR results, the Bayesian method assessed the support for the following five exclusive hypotheses: (1) no association with either trait; (2) association with trait 1 only; (3) association with trait 2 only; (4) both traits are associated, but distinct causal variants were for two traits; and (5) both traits are associated, and the same shares causal variant for both traits (*Foley et al., 2021*). The posterior probability is provided for each hypothesis (H0, H1, H2, H3, and H4). In this analysis, we set prior probabilities of the SNP being associated with trait 1 only (p1) at $1\times10^{-4}$; the probability of the SNP being associated with trait 2 only (p2) at $1\times10^{-4}$; and the probability of the SNP being associated with both traits (p12) at $1\times10^{-5}$. Two signals were considered to have strong evidence of colocalization if the posterior probability for shared causal variants (PH4) was $\geq0.8$. The analysis was performed in R software (4.3.3).

## Biological pathway analysis

The GENE2FUNC tool in FUMA (*Watanabe et al., 2017*) web application programming interface version 1.5.2 was used to investigate functional annotation and enrichment analysis of the genes coding for the 193 unique proteins identified by the PW-MR approach. This analysis involved calculating the log2 fold-change expression for each gene across 54 tissue types from the GTEx (genotype–tissue expression) database. We conducted gene enrichment analysis to identify overrepresented biological processes using the Gene Ontology (GO) database, which helped determine the biological processes most relevant to the genes identified. To investigate the involvement of the identified genes in functional and signaling pathways, we used the KEGG pathway database. Enrichment results were considered significant if they passed a 5% FDR $P$-value threshold. Additionally, only enrichments involving at least two overlapping genes with the gene sets were considered to ensure robustness and biological relevance.

## Druggability evaluation

The analysis of drug targets for the significant proteins identified in the MR analysis was conducted using data from OpenTargets (*Ochoa et al., 2023*) (v.22.11), which is publicly accessible. We selected all drugs for which there was evidence for an association with the protein of interest.

# Acknowledgements

The authors thank all the participants and investigators of the studies (PRACTICAL, FinnGen, deCODE, UKB-PPP) for providing the invaluable data used in this research. Their commitment and effort have been essential for the success of this study. Furthermore, special thanks to our colleagues and collaborators for their insightful discussions and feedback throughout the research process. This research was funded by Beijing Municipal Natural Science Foundation (grant no. JQ24059, no. L234038, no. L248076), and National Natural Science Foundation of China (grant no. 82274015).

# Additional information

## Funding

| Funder | Grant reference number | Author |
|---|---|---|
| Beijing Municipal Natural Science Foundation | JQ24059 | Xiaocong Pang |
| Beijing Municipal Natural Science Foundation | L234038 | Xiaocong Pang |

| Funder | Grant reference number | Author |
|---|---|---|
| Beijing Municipal Natural Science Foundation | L248076 | Zhuona Rong |
| National Natural Science Foundation of China | 82274015 | Xiaocong Pang |

The funders had no role in study design, data collection and interpretation, or the decision to submit the work for publication.

## Author contributions

Lin Chen, Data curation, Software, Visualization, Methodology, Writing – original draft; Yanlun Gu, Validation, Methodology; Yuke Chen, Wei Yu, Investigation, Methodology; Ying Zhou, Formal analysis, Supervision, Writing – review and editing; Zhuona Rong, Supervision, Funding acquisition, Visualization, Writing – review and editing; Xiaocong Pang, Resources, Supervision, Funding acquisition, Project administration, Writing – review and editing

## Author ORCIDs

Zhuona Rong ![ORCID] https://orcid.org/0000-0003-0636-9477
Xiaocong Pang ![ORCID] https://orcid.org/0000-0002-5951-5944

Reviewer #1 (Public review): https://doi.org/10.7554/eLife.101584.3.sa1
Reviewer #2 (Public review): https://doi.org/10.7554/eLife.101584.3.sa2
Reviewer #3 (Public review): https://doi.org/10.7554/eLife.101584.3.sa3
Author response https://doi.org/10.7554/eLife.101584.3.sa4

# Additional files

## Supplementary files

Supplementary file 1. Supplementary tables for the MR study on plasma proteins and PCa. (A) Results of genome-wide significant hits and assessment of heterogeneity in meta-analysis. (B) Cross-phenotype associations with PC from iCPAGdb. (C) Full results of the proteome-wide MR in deCODE including coloc and sensitivity analyses. (D) Full results of the proteome-wide MR in UKB_PPP including coloc and sensitivity analyses. (E) Results comparison between proteins significant in either dataset or in both. (F) OpenTargets drug targets.

MDAR checklist

## Data availability

All data used in this secondary analysis are publicly available. The GWAS summary statistics of blood pQTLs from deCODE can be found at https://www.decode.com/summarydata/, and those of the UKB-PPP can be found at https://www.ukbiobank.ac.uk/. Summary statistics of PCa from the PRAC-TICAL consortium can be obtained from OpenGWAS (https://gwas.mrcieu.ac.uk/), and those of the FinnGen study can be obtained from https://r10.finngen.fi/. The OpenTargets data were downloaded directly from the OpenTargets website (https://platform.opentargets.org/downloads/data).

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
