## [Editor Report · eLife Assessment]

This study presents a meta-analysis of two independent genome-wide association studies (GWAS) that investigate the role of plasma proteins as potential biomarkers for enhancing the early detection of prostate cancer (PCa). The results provide **useful** confirmatory data that support existing evidence currently published. The evidence is **incomplete**: the study does not provide a comprehensive synthesis of all currently published work, does not explore other clinical outcomes related to prostatic disease, and its findings have not been validated through an external cohort study. These shortcomings notwithstanding, the work may be of interest to researchers studying correlates and predictors of prostate cancer risk.

---

## [Referee Report · Reviewer #1 (Public review)]

Summary:

In Causal associations between plasma proteins and prostate cancer: a Proteome-Wide Mendelian Randomization the authors present a manuscript which seeks to identify novel markers for prostate cancer through analysis of large biobank-based datasets, and to extend this analysis to potential therapeutic targets for drugs. This is an area which is already extensively researched, but remains important, due to the high burden and mortality of prostate cancer globally.

Strengths:

The main strengths of the manuscript are the identification and use of large biobank data assets, which provide large numbers of cases and controls, essential for achieving statistical power. The databases used (deCODE, FinnGen and the UK Biobank) allow for robust numbers of cases and controls. The analytical method chosen, Mendelian Randomization, however, may not be appropriate to the problem (without extensive validation, MR can be prone to false or misleading discoveries). The manuscript also integrates multi-omic datasets, here using protein data as well as GWAS sources to integrate genomic and proteomic data.

Weaknesses:

The main weaknesses of the manuscript relate to the following areas:

(1) The failure of the study to analyse the data in the context of other closely related conditions such as benign prostatic hyperplasia (BPH) or lower urinary tract symptoms (LUTS), which have some pathways and biomarkers in common, such as inflammatory pathways (including complement) and specific markers such as KLK3. As a consequence, it is not possible for readers to know whether the findings are specific to prostate cancer, or whether they are generic to prostate dysfunction. Given the prevalence of prostate dysfunction (half of men reaching their sixth decade), the potential for false positives and overtreatment from non-specific biomarkers is a major problem, resulting in the evidence presented in this manuscript being weak. Other researchers have addressed this issue using the same data sources as presented here, for example in this paper looking at BPH in the UK Biobank population.

https://www.nature.com/articles/s41467-018-06920-9

(2) There is no discussion of Gleason scores with regard to either biomarkers or therapies, and a general lack of discussion around indolent disease as compared with more aggressive variants. These are crucial issues with regard to the triage and identification of genomically aggressive localized prostate cancers. See for example the work set out in: https://doi.org/10.1038/nature20788. In the revised version of the manuscript the authors set this out as a limitation, but this does not solve the core problem, which is that without this important biological context, the findings are unlikely to be robust.

(3) An additional issue is that the field of PCa research is fast-moving. The manuscript cites ~80 references, but too few of these are from recent studies and many important and relevant papers are not included. The manuscript would be much stronger if it compared and contrasted its findings with more recent studies of PCa biomarkers and targets, especially those concerned with multi-omics and those including BPH. In the latest revised version of the manuscript, some changes have been made, but the source data are still too limited for in-depth analysis.

(4) The Methods section provides no information on how the Controls were selected. There is no Table providing cohort data to allow the reader to know whether there were differences in age, BMI, ethnic grouping, social status or deprivation, or smoking status, between the Cases and Controls. These types of data are generally recorded in Biobank data; in the latest version of the manuscript the authors state that they don't have any ability to derive matched data, which again prevents deep analysis of the data.

Assessing impact:

Because of the weaknesses of the approach identified above, without further additions to the manuscript, the likely impact of the work on the field is minimal. There is no significant utility of the methods and data to the community, because the data are pre-existing and are not newly introduced to the community in this work, and mendelian randomization is a well-described approach in common use, and therefore the assets and methods described in the manuscript are not novel. In addition, Mendelian randomization is not always appropriate, especially when analysing publicly available data, see:

Stender et al. Lipids in Health and Disease (2024) 23:286

https://doi.org/10.1186/s12944-024-02284-w

With regard to the authors achieving their aims, without assessing specificity and without setting their findings in the context of the latest literature, the authors (and readers) cannot know or assess whether the biomarkers identified or the druggable targets will be useful in the clinic.

In conclusion, adding additional context and analysis to the manuscript would both help readers interpret and understand the work, and would also greatly enhance its significance. For example, the UK Biobank includes data on men with BPH / LUTS, as analysed in this paper, for example, https://doi.org/10.1038/s41467-018-06920-9. In the latest version of the manuscript and through the responses to earlier review comments, the authors explain why this has not been possible, but this naturally limits the value of the research.

---

## [Referee Report · Reviewer #2 (Public review)]

This is potentially interesting work, but the analyses are attempted in a rather scattergun way, with little evident critical thought. The structure of the work (Results before Methods) can work in some manuscripts, but it is not ideal here. The authors discuss results before we know anything about the underlying data that the results come from. It gives the impression that the authors regard data as a resource to be exploited, without really caring where the data comes from. The methods can provide meaningful insights if correctly used, but while I don't have reasons to doubt that the analyses were conducted correctly, findings are presented with little discussion or interpretation. No follow-up analyses are performed.

This is much improved but there remain some small concerns and one large concern:

Using numbering from the previous review:

(1) This looks better, but I still don't understand the claim in the text: "We found 5 genetic risk loci contained at least one SNP passing the genome-wide significance threshold of P {less than or equal to} 5×10−8". Far more gene regions appear to cross 10^-8 in Figure 1. What am I missing?

(6) I don't understand the authors' response here. Early detection is important, but MR is not the right tool to investigate biomarkers for early detection. Biomarkers for early detection do not have to be causal biomarkers. The authors replied to this point, but the manuscript was unchanged.

(7) Again, the authors still state "193 proteins were associated with PCa risk" even though they acknowledge that their analysis does not test whether proteins associate with PCa risk or not. When an error is pointed out, and you acknowledge it, please change the manuscript to correct the text. Otherwise, what is the peer review process for?

The large concern is that these analyses, while now better explained, are still the product of a semi-automated procedure. It is a good procedure, but the manuscript essentially takes public data from different sources and uses this to create a manuscript. Overall, I think there is enough novel synthesis to justify publication, but it is not automatic.

Strengths:

The data and methods used are state-of-the-art.

Weaknesses:

The reader will have to provide their own translational insight.

---

## [Referee Report · Reviewer #3 (Public review)]

Summary of concerns about the revised submission from the Reviewing Editor:

With respect to Originality of the work, in the last 18 months, there have been 38 publications on combined topics of: (i) UK Biobank data, (ii) Mendelian randomization, (iii) and prostate cancer. The authors should consider the literature addressing prostate cancer via Mendelian randomization--specifically those using the UK Biobank data--published from 2024 onwards. A proper and comprehensive synthesis of recent findings should be made, to allow readers to compare and contrast how the work supports (or differs) from the findings presented in these other published studies.

With respect to the significance of the findings, we feel the study data are incomplete for the strength of evidence. Given the deluge of manuscripts and publications on similar topics, the study offers incremental evidence and lacks a synthesis of all currently published findings.

---

## [Author Response]

The following is the authors’ response to the original reviews.

**Reviewer #1 (Public review):**
Summary:In Causal associations between plasma proteins and prostate cancer: a Proteome-Wide Mendelian Randomization, the authors present a manuscript which seeks to identify novel markers for prostate cancer through analysis of large biobank-based datasets and to extend this analysis to potential therapeutic targets for drugs. This is an area that is already extensively researched, but remains important, due to the high burden and mortality of prostate cancer globally.Strengths:The main strengths of the manuscript are the identification and use of large biobank data assets, which provide large numbers of cases and controls, essential for achieving statistical power. The databases used (deCODE, FinnGen, and the UK Biobank) allow for robust numbers of cases and controls. The analytical method chosen, Mendelian Randomization, is appropriate to the problem. Another strength is the integration of multi-omic datasets, here using protein data as well as GWAS sources to integrate genomic and proteomic data.

Thank you for your positive feedback regarding the overall quality of our work and we greatly appreciate you taking time and making effort in reviewing our manuscript.

Weaknesses:The main weaknesses of the manuscript relate to the following areas:(1) The failure of the study to analyse the data in the context of other closely related conditions such as benign prostatic hyperplasia (BPH) or lower urinary tract symptoms (LUTS), which have some pathways and biomarkers in common, such as inflammatory pathways (including complement) and specific markers such as KLK3. As a consequence, it is not possible for readers to know whether the findings are specific to prostate cancer or whether they are generic to prostate dysfunction. Given the prevalence of prostate dysfunction (half of men reaching their sixth decade), the potential for false positives and overtreatment from non-specific biomarkers is a major problem, resulting in the evidence presented in this manuscript being weak. Other researchers have addressed this issue using the same data sources as presented here, for example, in this paper, looking at BPH in the UK Biobank population. https://www.nature.com/articles/s41467-018-06920-9

Thank you for your valuable comment. We fully agree that biomarker development must prioritize specificity to avoid overtreatment. While our study is a foundational step toward identifying potential therapeutic targets or complementary biomarkers for prostate cancer—not as a direct endorsement of these proteins for standalone clinical diagnosis. Mendelian randomization analysis strengthens causal inference by design, and we further ensured robustness through sensitivity analyses (e.g., MR-Egger regression for pleiotropy, Bonferroni correction for multiple testing). These methods distinguish true causal effects from nonspecific associations. Importantly, while PSA’s lack of specificity is widely recognized, its role in reducing PCa mortality underscores the value of biomarker-driven screening. Our findings align with the need to integrate multiple markers (e.g. combining a novel protein with PSA) to improve diagnostic precision. Translating these causal insights into clinical tools remains challenging but represents a necessary next step, and we emphasize that this work provides a rigorous starting point for future validation studies.

(2) There is no discussion of Gleason scores with regard to either biomarkers or therapies, and a general lack of discussion around indolent disease as compared with more aggressive variants. These are crucial issues with regard to the triage and identification of genomically aggressive localized prostate cancers. See, for example, the work set out in: https://doi.org/10.1038/nature20788

Thank you for pointing this out. We acknowledge that our original analysis did not directly address this critical issue due to a key data limitation: the publicly available GWAS summary statistics for PCa (from openGWAS and FinnGen) do not provide genetic associations stratified by phenotypic severity or molecular subtypes. This limitation precluded MR analysis of proteins specifically linked to aggressive disease. To partially bridge this gap, we integrate evidence from recent studies in the revised Discussion section to explore the relevance of potential biomarkers to aggressive PCa.

(3) An additional issue is that the field of PCa research is fast-moving. The manuscript cites ~80 references, but too few of these are from recent studies, and many important and relevant papers are not included. The manuscript would be much stronger if it compared and contrasted its findings with more recent studies of PCa biomarkers and targets, especially those concerned with multi-omics and those including BPH.

Thank you for your professional comments. We have rigorously updated the manuscript to include more recent publications and we systematically compare and contrast our findings with these recent studies in the revised Discussion section.

(4) The Methods section provides no information on how the Controls were selected. There is no Table providing cohort data to allow the reader to know whether there were differences in age, BMI, ethnic grouping, social status or deprivation, or smoking status, between the Cases and Controls. These types of data are generally recorded in Biobank data, so this sort of analysis should be possible, or if not, the authors' inability to construct an appropriately matched set of Controls should be discussed as a Limitation.

We thank the reviewer for raising this important methodological concern. We have expanded the Limitations section to state it.

“Lastly, our analysis relied exclusively on publicly available GWAS summary statistics from openGWAS and FinnGen, which did not provide individual-level data on covariates, resulting in no direct assessment of demographic or clinical differences between cases and controls.”

**Reviewer #2 (Public review):**
This is potentially interesting work, but the analyses are attempted in a rather scattergun way, with little evident critical thought. The structure of the work (Results before Methods) can work in some manuscripts, but it is not ideal here. The authors discuss results before we know anything about the underlying data that the results come from. It gives the impression that the authors regard data as a resource to be exploited, without really caring where the data comes from. The methods can provide meaningful insights if correctly used, but while I don't have reasons to doubt that the analyses were conducted correctly, findings are presented with little discussion or interpretation. No follow-up analyses are performed.In summary, there are likely some gems here, but the whole manuscript is essentially the output from an analytic pipeline.

We thank the reviewer for the thoughtful evaluation of our work. In response to the concerns regarding manuscript structure and interpretative depth, we have restructured the manuscript to present the Methods section before Results, ensuring transparency in data sources and analytical workflows. Additionally, the Discussion section has been substantially revised to provide mechanistic explanations for key findings (e.g., associated phenotype, causal proteins, druggable targets), contextualize results within recent multi-omics studies and highlight clinical implications. These revisions aim to transform the work from a pipeline-driven analysis to a biologically grounded investigation, offering actionable insights into prostate cancer pathogenesis and therapeutic development.

Taking the researchers aims in turn:(1) Meta-GWAS - while combining two datasets together can provide additional insights, the contribution of this analysis above existing GWAS is not clear. The PRACTICAL consortium has already reported the GWAS of 70% of these data. What additional value does this analysis provide? (Likely some, but it's not clear from the text.) Also, the presentation of results is unclear - authors state that only 5 gene regions contained variants at p<5x10-8, but Figure 1 shows dozens of hits above 5x10-8. Also, the red line in Figure 1 (supposedly at 5x10-8) is misplaced.

Thank you very much for your feedback. Although the PRACTICAL consortium constituted the majority of PCa GWAS data, our meta-analysis integrating FinnGen data enhanced statistical power enabling robust detection of low-frequency variants with minor allele frequencies. Moreover, FinnGen's Finnish ancestry (genetic isolate) helps distinguish population-specific effects. The presentation of results showed the top 5 gene regions contained variants at p < 5×10⁻⁸. We apologize for not noticing that the red line was not displayed correctly in the original figures included in the manuscript. We have updated it in the revised manuscript.

(2) Cross-phenotype analysis. It is not really clear what this analysis is, or why it is done. What is the iCPAGdb? A database? A statistical method? Why would we want to know cross-phenotype associations? What even are these? It seems that the authors have taken data from an online resource and have written a paragraph based on this existing data with little added value.

We appreciate the opportunity to clarify this analysis. The cross-phenotype analysis was designed to systematically identify phenotypic traits that share genetic or molecular pathways with prostate cancer, thereby uncovering pleiotropic mechanisms or shared risk factors. Here, iCPAGdb (integrated Cross-Phenotype Association Genetics Database) is a curated repository that aggregates GWAS summary statistics and evaluates genetic pleiotropy using LD-proxy associations from the NHGRI-EBI GWAS Catalog. Prostate carcinogenesis involves multisystem interactions, including spanning endocrine dysregulation, immune microenvironment remodeling and metabolic reprogramming, rather than isolated molecular pathway disruptions. Therefore, it is indispensable for discriminating primary pathogenic drivers from secondary compensatory responses, ultimately informing the development of precision therapeutic strategies.

In response to your concerns, we have revised the Results section to explicitly define the rationale and methodology of cross-phenotype analysis and restructured the Discussion to interpret phenotype-PCa associations within unified biological frameworks (e.g., metabolic dysregulation, androgen signaling), rather than presenting them as isolated findings.

(3) PW-MR. I can see the value of this work, but many details are unclear. Was this a two-sample MR using PRACTICAL + FinnGen data for the outcome? How many variants were used in key analyses? Again, the description of results is sparse and gives little added value.

We thank you for raising this issue. Two-sample MR refers to an analytical design where genetic instruments for the exposure (plasma proteins) and genetic associations with the outcome (PCa) are derived from non-overlapping populations. This ensures complete sample independence between exposure and outcome datasets to avoid confounding biases, regardless of whether the outcome data originate from single or multiple cohorts. The meta-analysis of PRACTICAL and FinnGen GWAS generates 27,210 quality-controlled variants (p < 5×10⁻⁸, MAF ≥ 1%, LD-clumped r² < 0.1) used in key analyses. Regarding the concern about sparse interpretation, we have substantially expanded the Discussion by comparing significant protein findings (e.g., MSMB, SERPINA3) with results from existing functional studies and multi-omics datasets and unravelling new insights.

(4) Colocalization - seems clear to me.(5) Additional post-GWAS analyses (pathway + druggability) - again, the analyses seem to be performed appropriately, although little additional insight other than the reporting of output from the methods.

The post-MR druggability and pathway analyses serve two primary scientific purposes: (1) therapeutic prioritization - systematically evaluating which MR-identified proteins represent tractable drug targets (either through existing FDA-approved agents or compounds in clinical development) with direct relevance to cancer or PCa management, and (2) mechanistic hypothesis generation - mapping these candidate proteins to coherent biological pathways to guide future functional validation studies investigating their causal roles in prostate carcinogenesis. In response to your feedback, we have restructured the Discussion section under the subheading “Biological Mechanisms and Druggable Targets” to synthesize these findings, explicitly linking biological pathway to therapeutic targets.

Minor points:(6) The stated motivation for this work is "early detection". But causality isn't necessary for early detection. If the authors are interested in early detection, other analysis approaches are more appropriate.

We appreciate your insightful feedback. Early detection is one motivation for this work, meanwhile, our goal is also to identify causally implicated proteins that may serve as intervention targets for PCa prevention or therapy. Establishing causality is critical for distinguishing biomarkers that drive disease pathogenesis from those that are secondary to disease progression, as the former holds greater specificity for early detection and prioritization of therapeutic targets. While we acknowledge that validation for early detection may require additional methodologies, MR analysis provides a foundational step by prioritizing candidate proteins with causal links to disease. This approach ensures that downstream efforts focus on biomarkers and targets with the greatest potential to alter disease trajectories, rather than merely correlative markers.

(7) The authors state "193 proteins were associated with PCa risk", but they are looking at MR results - these analyses test for disease associations of genetically-predicted levels of proteins, not proteins themselves.

True, in MR, the exposure of interest is the lifelong effect of genetically predicted protein levels. This approach is designed to infer causality while avoiding confounding and reverse causation, as genetic variants are fixed at conception and unaffected by disease processes. When we state “193 proteins were associated with PCa risk,” we specifically refer to proteins whose genetically predicted levels (based on instrument SNPs from protein QTLs) show causal links to PCa. Importantly, MR does not measure the direct association between observed protein concentrations and disease. Instead, it estimates the lifelong causal effect of protein levels predicted by genetics. This distinction is critical for disentangling cause from consequence. For example, a protein elevated due to tumor progression would not be identified as causal in MR if its genetic predictors are unrelated to PCa risk.

We acknowledge that clinical translation requires further validation of these proteins in observational studies measuring actual protein levels. However, MR provides a robust first step by prioritizing candidates with causal roles, thereby reducing the risk of investing in biomarkers confounded by disease processes.

**Reviewer #1 (Recommendations for the authors):**
As outlined above, the major weakness of the manuscript is its failure to consider BPH / LUTS, and whether the markers and targets are specific to PCa or not. Specific improvements that the authors could consider might include a literature review of the features identified for their 20 high-risk proteins, and ideally also analyze whether these proteins are upregulated or downregulated in the databases they have analysed (for example it will be easy to analyze whether these proteins are dysregulated in BPH patients as these are specifically identified in the UK Biobank).The authors may be able to gain context for this approach by looking at papers analyzing BPH and the complement cascade and other proteins from the authors' top 10 or top 20, for example: https://doi.org/10.1002/pros.24639IF : 2.6 Q2Other sources can be identified by examining the literature for recent omics papers analysing BPH, especially those that analyse and compare BPH / PCa specifically.

Thank you for highlighting the critical need to distinguish PCa-specific biomarkers from those shared with BPH. In response, we conducted a literature review of multi-omics datasets and prospective cohort studies, systematically evaluating the specificity of prioritized proteins by comparing their expression trends in PCa and BPH or benign prostate tissues. These findings are now integrated into the revised Discussion section under the subheading " Plasma Proteins Causal Links to Prostate Cancer".

In the Discussion, the paragraph (line 288) on PSA is extremely weak. The authors state that further research is needed, and yet only reference four articles (from 2008, 2010, 2012, 2014), none of which are from the last decade. Considerable amounts of research from the last ten years have been published on PSA, for example, see this article from 2018, which specifically analyses PSA in the context of the UK Biobank. This section should be made more up-to-date with the latest literature findings. https://doi.org/10.1038/s41467-018-06920-9

Thank you very much for your feedback. We acknowledge the need to strengthen the discussion on PSA by incorporating recent literature. In the revised manuscript, we have expanded the PSA discussion to integrate contemporary research on the prognostic role of PSA in the progression of PCa and its limitations in cancer screening, ensuring that our discussion reflected the current consensus and controversies.

Also in the Discussion, the analysis of phenotypic indicators is insufficiently comprehensive and should reference other recent research. For example, this recent UK Biobank study dealt with a wide range of conditions, including prostate cancer, and identified similar factors to those identified in this paper. The authors should compare and contrast their phenotypic findings with the existing literature. https://doi.org/10.1038/s41588-024-01898-1

Thank you for addressing the comprehensiveness of phenotypic analysis. We have learned recent large-scale phenome-wide analyses (linked in your feedback) that explore multi-omics biomarkers and their associations with a range of different diseases. We have compared and contrasted our phenotypic findings with the existing literature and revised the Discussion section to interpret phenotype-PCa associations, emphasizing both shared pathways and disease-specific signals.

Under Methods, there is too little information on how Controls were selected, whether any matching process was conducted, or whether there are fundamental differences between the cases and controls (such as smoking status, BMI, comorbidities). The authors use R, and a library such as MatchIt could be used to ensure that the Controls cohort is appropriately matched to the Cases.

As outlined above, we acknowledge that our original analysis did not directly address this critical issue due to a key data limitation. The publicly available GWAS summary statistics for PCa (from openGWAS and FinnGen) do not provide individual-level data on covariates, resulting in no direct assessment of demographic or clinical differences between cases and controls.

An important final point is that, as far as I can tell, no UK Biobank Application Number has been specified in the manuscript. This is vital to establish that there was an original hypothesis being investigated (as opposed to data dredging of open access resources), especially in light of the largely mechanistic flow of the manuscript and lack of PCa and relevant confounder-specific discussion. The authors may be aware of the work of Stender et al (2024) regarding formulaic papers using Mendelian randomization, especially that "[All] combinations of exposure and outcome results based on data available in IEU openGWAS (https://gwas.mrcieu.ac.uk/) can be browsed online on epigraphDB.org. In other words, these results are, in effect, already published. Reporting them again in a scientific paper adds nothing to what can be looked up online in minutes." The authors may wish to address this issue directly.Stender, S., Gellert-Kristensen, H. & Smith, G.D. Reclaiming Mendelian randomization from the deluge of papers and misleading findings. Lipids Health Dis 23, 286 (2024). https://doi.org/10.1186/s12944-024-02284-w

We confirm that all data used in this study were obtained from publicly available GWAS summary statistics (e.g., PRACTICAL consortium, FinnGen) and proteomic datasets (deCODE, UKB-PPP). Our research was guided by a predefined hypothesis to investigate causal plasma protein biomarkers for prostate cancer, rather than exploratory data mining. The analytical pipelines and integrative approaches (e.g., colocalization, druggability assessment) were specifically designed to address this hypothesis, aligning with the ethical use of open-access resources.

**Reviewer #2 (Recommendations for the authors):**
There are several specific recommendations in the public review (e.g., clarify the contribution of the GWAS). Otherwise, there is nothing clearly incorrect, but translational insight is missing - the analyses are not clearly connected to the scientific literature. This is a limitation rather than a flaw - the manuscript will likely still be useful to readers.

We thank you for highlighting the need to strengthen translational insights and contextualize our findings within existing literature. In the revised manuscript, we have expanded the Discussion section to systematically compare our results with prior mechanistic and clinical studies, including the shared pathways of associated phenotypes, the potential of significant proteins in biomarkers and therapeutic targeting. These revisions ensure our analyses are firmly rooted in the scientific literature.